# Secular Trends in Systemic Sclerosis Mortality in the United States from 1981 to 2020

**DOI:** 10.3390/ijerph192215088

**Published:** 2022-11-16

**Authors:** Jing-Xing Li

**Affiliations:** 1Department of General Medicine, China Medical University Hospital, No. 2, Yude Rd., North District, Taichung 404327, Taiwan; jamesli.0531@gmail.com; Tel.: +886-905-770-531; 2School of Medicine, China Medical University, Taichung 404327, Taiwan; 3Graduate Institute of Clinical Laboratory Sciences and Medical Biotechnology, National Taiwan University, Taipei 100225, Taiwan

**Keywords:** systemic sclerosis, systemic scleroderma, mortality, underlying cause of death, multiple causes of death, age-standardized mortality rate

## Abstract

Background: Systemic sclerosis (SSc) has the highest mortality rate among autoimmune disorders. Individuals with SSc frequently die from complications or infections related to SSc. Nonetheless, the sex–age–period interaction of SSc is complex and remains unclear. The study aims to analyze the secular trend of SSc mortality based on data regarding underlying cause of death (UCD) and multiple causes of death (MCD) and clarify the sex–age interaction with time. Methods: The multiple-cause mortality statistics provided by the National Center for Health Statistics were used to identify all deaths in the United States from 1981 to 2020 in which SSc was indicated anywhere on the death certificates. The age-standardized mortality rate (ASMR) was determined for both sexes, as well as the variations in these rates. Joinpoint regression analysis was utilized to determine the annual percentage change (APC) of ASMR. Results: A total of 44,672 and 66,259 individuals who died between 1981 and 2020 were identified based on the UCD and MCD data, respectively. According to the UCD data, SSc-related AMSR (SSc-ASMR) of the male and female decedents, respectively, declined from 5.01 and 1.94 in 1981–1990 to 4.77 and 1.32 in 2011–2020, respectively (mortality rate ratio 0.95, 95% confidence interval 0.92–0.98). From 1986 to 1999, the APC of SSc-ASMR in female decedents decreased except for those aged 45–64 years (APC 2.1%, *p* = 0.002). For MCD analysis, in trend 1, only APC of SSc-ASMR in male decedents aged 45–64 years decreased. The SSc-ASMR of both male and female decedents fell on trend 2 arm. In 2011–2020, the ratio of UCD to MCD increased across all age groups for both sexes compared to 1981–1990. Overall, compared to the male decedents, the SSc-ASMR in female decedents increased significantly before 1999, peaked in 1999, followed by continuous decrease until 2020 according to UCD and MCD statistics. Conclusions: Over the past four decades, the SSc deaths based on the MCD data were 1.48 times more than the UCD data, and the proportion of UCD over MCD increased over time. The SSc-ASMRs in all the sex–age groups significantly decreased over the past two decades. Notably, the mortality rate ratio of women to men with SSc increased in the past four decades.

## 1. Introduction

Of all rheumatic diseases, systemic sclerosis (systemic scleroderma; SSc) has the highest case-specific mortality and complications, despite evidence of improved survival. SSc is a rare, highly morbid autoimmune connective tissue disease, distinguished by a wide variety of clinical manifestations in multiple organ systems and characterized by chronic and progressive tissue and organ fibrosis [1]. From 2000 to 2014, North America was the region with the highest SSc mortality rate worldwide [2]. Various research groups have conducted retrospective cohort studies and meta-analyses of the SSc literature [3,4,5,6] throughout the past two decades, producing a variety of mortality estimates and risk factor assessments. Several meta-analyses revealed that the standardized mortality rate (SMR) for SSc has not decreased over time given that the overall survival rates of the reference populations have improved.

SSc is associated with a 2.5-fold higher risk of mortality than the general population, and there is no evidence that survival has improved with time [7]. Investigating SSc mortality is difficult due to the low incidence rate of 25 persons per million per year, which has not changed over the past four decades, and the small number of deaths in each study, which precluded further analysis by sex and age. Using mortality data compiled by the National Vital Statistics System affiliated with the National Center for Health Statistics (NCHS), a previous study identified 46,798 SSc deaths from 1968 to 2015 in the United States and reported increases in the annual age-standardized mortality rate (ASMR) of SSc from 1968 to 2000, followed by a steady decline from 2001 to 2015, but there was no further analysis stratified by age groups [5]. In light of the recent development of hematopoietic stem cell transplant and monoclonal antibody treatments for SSc over the past decade, an updated review of mortality seems essential.

This research aims to determine whether SSc mortality changed in the past four decades and examine the secular trends of SSc mortality in the United States from 1981 to 2020, based on underlying cause of death (UCD) data on the death certificates. Because the ratio of SSc mortality to non-SSc mortality remains high, the disease burden may be underestimated in analyses of SSc mortality based solely on the UCD. Consequently, multiple causes of death (MCD) will also be investigated. 

## 2. Methods

### 2.1. Data Sources 

We used the multiple-cause mortality files in the NCHS database to identify all deaths in the United States from 1981 to 2020 in which SSc was mentioned on the death certificates. The MCD data from the NCHS accounted for up to 20 causes of death documented by medical certifiers on each death certificate. In addition to demographic information (sex and age), the MCD statistics also include information on the decedent’s place of residence and place of death. More than 99% of deaths among the United States residents in all 50 states and the District of Columbia are captured by the database. Data on SSc deaths was retrieved using the Wide-Ranging Online Data for Epidemiologic Research (WONDER) web program from the Centers for Disease Control and Prevention (CDC). Each death record contains one UCD, up to 20 additional MCDs, and demographic information. Trends in SSc-related mortality were defined as deaths having International Classification of Diseases, Tenth Revision, Clinical Modification (ICD-10-CM) codes M34.0, M34.1, M34.8, or M34.9 and International Classification of Diseases, Ninth Revision, Clinical Modification (ICD-9-CM) code 710.0 listed among any of the 20 MCDs or as the UCD. ICD-10-CM and ICD-9-CM codes were used for identifying SSc in 2011–2020 and 1979–2015, respectively.

### 2.2. Mortality Rate

This study will involve two definitions of death: UCD and MCD. The definitions were derived from the ICD-10 Instruction Manual [8]. UCD refers to the disease or injury that launched the sequence of morbid events that led directly to death, or the accident or act of violence that caused the fatal injury. Otherwise, MCD comprises not only the UCD, but also the immediate cause of death (Appendix A) and any intermediate and contributing conditions stated on the death certificates. According to the UCD and MCD data, the number of deaths and ASMR were classified (deaths per one million people) into four age categories (0–44, 45–64, 65–74, and ≥75 years old) by sex for 1981–1990 and 2011–2020. The mortality rate for all ages was adjusted for age using the population of the United States in 2000 as a reference.

### 2.3. Statistical Analysis

To determine the extent of change over the past four decades, the mortality rate ratio (RR) of SSc deaths was calculated in 2011–2020 relative to those in 1981–1990. An RR less than 1 indicates a decrease in the mortality rate, and the lower the RR, the larger the decreases in mortality rates. Using joinpoint regression analysis, the annual percentage change (APC) of ASMR was then determined. Utilizing the findings of the joinpoint regression analysis, any statistically significant annual fluctuations in mortality over time were identified.

## 3. Results

A total of 44,672 and 66,259 individuals who died of SSc between 1981 and 2020 were identified according to the UCD (Appendix A) and MCD (Appendix A) data, respectively. The SSc deaths based on MCD data were 1.48 times higher than UCD data in 1981–2020. 

### 3.1. Trend of SSc-ASMR

Figure 1 plots the fluctuation of ASMR, which was illustrated in Appendix A, in strata of sex and age. According to UCD statistics, the overall incidence of SSc-related ASMR (SSc-ASMR) in female decedents decreased from 1981 to 1986, rose significantly between 1986 and 1999, and peaked in 1999 (7.03 per million), before declining steadily until 2020 (Figure 1A). Alternately, SSc-ASMR in male decedents remained largely steady with a slight increase from 1981 to 2001, followed by a slight drop until 2020 (Figure 1A). SSc-ASMR in females aged 45–64 years declined from 1981 to 1986 (Figure 1E). In addition, SSc-ASMR in females aged ≥75 years declined slightly from 30.19 to 28.88 per million between 2001 and 2013 (Figure 1I). While for MCD, the overall SSc-ASMR of female decedents increased from 1981 to 1999 and peaked in 1999, followed by steady decline until 2020 (Figure 1B). Male SSc-ASMR decreased throughout the follow-up period. 

### 3.2. UCD of SSc

Table 1 demonstrates the number of deaths, mortality rates, and mortality RRs of 2011–2020/1981–1990 according to the UCD and MCD data, stratified by sex and age. According to UCD data, the percentage of decedents aged 45–64 years increased from 38.3% for male decedents and 45.2% for female decedents in 1981–1990 to 31.8% and 42.5%, respectively, in 2011–2020. The SSc-AMSR of the male and female decedents declined from 1.94 and 5.01, respectively, in 1981–1990 to 1.32 and 4.77, respectively, in 2011–2020. The overall RR for the male decedents was 0.68 (95% CI 0.64–0.72), and it was 0.95 (95% CI 0.92–0.98) for the female decedents. The decrease of RR was more pronounced among male decedents aged 45–64 years and female decedents aged 0–44 years. Among all sex–age groups, the SSc-ASMR of the female decedents aged ≥75 years markedly elevated in 2011–2020 (RR 1.66, 95% CI 1.55–1.77). 

Appendix A reveals the APC of SSc-ASMR which was estimated through the joinpoint regression analysis. Regarding the annual mortality trends according to the UCD data, the APC for the male decedents was 0.4% during trend 1 (1981–2002), and the APC in the age range of 0–44 was significantly increased (*p* = 0.019). In trend 2, the APC in female decedents fell dramatically throughout all age groups. In terms of female decedents, only the APC in the age group 45–64 dropped considerably, whereas the APC in all other age groups increased significantly. For males of all ages, the APCs decreased significantly in trend 2. On trend 2 arm, the SSc-ASMR of female decedents declined dramatically in the 0–44 and 65–74 age groups but increased significantly in the 45–64 age group. In trends 3 and 4, SSc-ASMR dropped substantially. A suboptimal change of APC in females aged ≥75 years was detected during trend 1 (*p* = 0.053).

### 3.3. MCD of SSc 

Patients with SSc most frequently died from complications such as interstitial lung disease (ILD) and pulmonary arterial hypertension (PAH) without appropriate therapeutic control. Table 2 lists the general management for SSc according to the latest European League against Rheumatism (EULAR) recommendations released in 2016 [9]. The principle of SSc treatment is based on the organ involved and tailored to individuals. 

According to the MCD data, the SSc-ASMR was higher in women (7.98 per million) than in men (3.09 per million) in 1981–1990, but both decreased to 6.77 and 1.86 in 2011–2020, with RRs of 0.85 (95% CI 0.83–0.87) for female decedents and 0.60 (95% CI 0.57–0.63) for male decedents. The annual mortality trends according to the MCD data followed the same pattern as those according to the UCD data. Female decedents aged ≥75 years had the highest RR among all sex–age groups (RR 1.33, 95% CI 1.26–1.40). The SSc-ASMR for female decedents of all ages decreased dramatically. During trend 1, only APC of SSc-ASMR in males aged 45–64 years decreased (Appendix A). A suboptimal change of APC in males aged 0–44 years was shown in trend 1 (*p* = 0.051). The SSc-ASMR of both male and female decedents fell on trend 2 arm. 

### 3.4. Ratio of UCD to MCD

According to UCD and MCD statistics, the age distribution of SSc-affected decedents was comparable. To investigate the weight of UCD during different time periods, subgroup analysis was also performed with age and sex stratifications. Figure 2 presents the proportion of UCD over MCD. In 2011–2020, the ratio of UCD to MCD increased across all age groups for both sexes compared to 1981–1990. The overall UCD/MCD ratio increased from 62.6% (8,067/12,896, 1981–1990) to 70.3% (11,934/16,965, 2011–2020) (Table 1). As individuals’ age increased, the UCD/MCD ratio decreased. Appendix A shows mortality rate based on UCD and MCD data stratified by age group of 5 years.

## 4. Discussion

In the present study, 1.48 times as many SSc deaths were detected in MCD data compared to the UCD data. In each age group, the SSc-ASMRs for both sexes have dropped over the past two decades. Comparing the 1981–1990 group to the 2011–2020 group, the SSc-ASMR among female decedents aged ≥75 years increased significantly from 15.68 to 26.00, with an RR of 1.66 (95% CI 1.55–1.77). The most notable conclusion was that SSc-ASMR decreased by around 3.3% and 4.4% annually in male and female decedents from 2002 to 2020 and 2013 to 2020, respectively.

This study found an increasing trend of SSc-ASMR in women from 1981 to 1999, followed by a reduction from 1999 to 2020, according to UCD and MCD data. However, the SSc-ASMR in men remained steady until 2001, followed by a gradual reduction to the present. The finding was consistent with other population-based cohort studies in the United States [5,6,7]. A French population-based study likewise revealed a drop in the SMR for both sexes from 2000 to 2011 [12].

A sex–age–period interaction was clearly demonstrated in the current study. Among the female decedents aged 0–74 years, the SSc-ASMR was lower in 2011–2020 than it was in 1981–1990. However, the SSc-ASMR were much higher in 2011–2020 compared to 1981–1990 among the decedents aged ≥75 years. Increasing age has historically been related to increased SSc mortality [13]. In line with this, the findings of this study indicated that SSc-ASMR increased with age. Two recent meta-analyses [4,14] found that older age at diagnosis was associated with a higher unadjusted mortality risk. However, in their meta-regression analyses, Elhai et al. [3] revealed that age at enrollment was not associated with SMR, and there was no significant association of sex and SMR per meta-regression over time. Sex-specific meta-analysis revealed a significant increase in overall SMR for both sexes, whereas no difference in the SMR between men and women was reported [15]. Conversely, male sex was associated with significantly increased risk for death in these meta-analyses [4,14,15].

According to the trendline in Figure 1, in 1981, the overall SSc-ASMRs of female to male decedents were 2.58 (4.90/1.90) and 2.42 (7.45/3.08) in UCD and MCD analysis, respectively. The mortality RR in 2020 was 3.53 (3.95/1.12) and 3.60 (6.04/1.68) based on UCD and MCD data, respectively. The mortality RR of women to men with SSc rose in the past four decades. It seemed that men diagnosed with SSc benefited more from the improvement of the management for SSc-related complications than women diagnosed with SSc. 

Apart from the disparity in sex, race and geographic region disparities were also demonstrated. Rodriguez-Pla et al. [6] showed that the highest SSc-ASMR was in African Americans, at 5.70 per million, followed by American Indians or Alaska Natives. Per geographic region, the lowest SSc-ASMR was 2.25 per million in Hawaii, whereas the highest one was 6.29 per million in South Dakota.

A notable finding of this study was the nonsignificant decline in SSc mortality among male decedents aged 45–64 years and the nonsignificant increase in SSc mortality, respectively, aged 65–74 years and ≥75 years according to the UCD data during trend 1. Additionally, the overall APC of SSc-ASMR was unremarkable, resulting from the uneven changes in the death rates among age groups. For female decedents, overall APC of SSc-ASMR was also nonsignificant, although significant changes in each age group were observed in trend 1 (1981–1986). Similar effects were also identified in the MCD analysis, but exclusively for men; the overall APC of SSc-ASMR in male decedents was not remarkable in trend 1 (1981–2000). Additional research is required to investigate the underlying mechanisms behind the impact of sex–age–period interaction on SSc-ASMRs.

The years 1999 and 2002 marked the turning points for ASMR in females and males, respectively. After this time period, ASMR declined, with females experiencing a faster decline. The introduction of endothelin 1 receptor antagonists and phosphodiesterase type 5 inhibitor for PAH at the beginning of the twenty-first century has significantly improved pulmonary hemodynamics and clinical outcomes [16,17]. It is proposed that scleroderma renal crisis treated with an angiotensin-converting enzyme inhibitor had better outcomes [18,19]. Afterwards, cyclophosphamide was developed and became the cornerstone of treatment for ILD associated with SSc. These medications significantly reduced mortality resulting from pulmonary complications associated with SSc. Rituximab, a CD20 monoclonal antibody, has been demonstrated to ameliorate skin fibrosis and prevent lung fibrosis from deteriorating in SSc patients [20]. The present study found the UCD of SSc in female decedents decreased significantly, which may be attributable to these potential strategies for the dismal prognosis of PAH. 

SSc-related ILD (SSc-ILD) is associated with early mortality. In a study from a single location in the United States, the proportion of deaths attributable to pulmonary fibrosis increased from 6% in 1972–1976 to 33% in 1997–2001, making ILD the most common cause of mortality associated with SSc [21]. Pulmonary hypertension is a common complication of ILD. Individuals with SSc-related PAH (SSc-PAH) have a mortality risk that is threefold higher than the individuals with the idiopathic PAH [22]. In the United States, SSc-*PAH* mortality stayed steady from 2003 to 2008, then declined by 3% annually from 2008 to 2016 [23]. Furthermore, a meta-analysis revealed that SSc-PAH was significantly associated with a greater pooled risk of mortality than SSc without PAH (RR 3.12, 95% Cl 2.44–3.98) [24]. Eighteen studies comprising a total of 12,829 individuals reported the following causes of death: 19.7% cardiac disease, 16.8% ILD, 13.1% pulmonary hypertension, and 13.8% renal disease [25]. Furthermore, Elhai et al. [3] reported that cardiac involvement was more frequent than pulmonary involvement.

In trend 1, APC of SSc-ASMR significantly decreased in the UCD analysis of female decedents aged 45–64 years (*p* = 0.036). As suggested by M. Nikpour et al. [26], early diagnosis and treatment of SSc may have contributed significantly to the drop in SSc mortality in recent decades. Given the vast clinical complexity of SSc, there is no standard treatment for SSc and management techniques are highly heterogeneous and personalized. The accessibility of and adherence to these treatments may have been relatively low among the individuals aged 45–64 years.

Of note, the use of the MCD data could result in conclusions that are different from those drawn using the UCD data. According to the UCD data, SSc-ASMR among male decedents aged 0–44 years significantly increased from 1981 to 2002, although the MCD data did not suggest any significant changes. Nevertheless, there was no significant change among male decedents aged 45–64 years from 1981 to 2004 according to UCD data, but a significant change was discovered in the same age group based on MCD analysis (*p* < 0.001). 

In the period between 1981 and 1990, SSc was selected as the UCD in 62.6% of MCD cases. As the age of the deceased increased, this proportion declined. In 1981–1990, among male decedents aged 0–44 and ≥75 years, the UCD/MCD ratio was, respectively, 73.4% and 55.5%, but it increased to 76.6% and 67.1% in 2011–2020. For both sexes, the proportion of death in SSc attributed to UCD/MCD ratio decreased steadily with increasing age. However, the ratio rose slightly among males aged 65–74 years, from 71.3% to 71.8%. Ratanawatkul et al. [23] indicated a similar pattern using UCD and MCD data. Among patients with SSc, the risk of dying from cancer, cardiopulmonary complications, infections, or malignancies increases with age; consequently, SSc is less likely to be selected as the UCD in such cases. 

According to UCD and MCD statistics, SSc-ASMR decreased by 11.8% and 21.9% in the past four decades. Similar trends in SSc mortality in the United States were observed in the present analysis as in prior observational studies [5,7,27]. Nevertheless, according to a recent meta-analysis, there was a trend for a non-significant fall in SMR during meta-regression [15]. This result should be carefully interpreted not just based on *p* value, because heterogeneity in clinical characteristics, such as sex, age, race, study design, severity of organ impairment, and proportion of individuals with cardiac, pulmonary, and renal involvement, tended to confound meta-analysis and seeded bias in study results. 

### Strengths and Limitations

The strength of the study is the use of a systemic approach to access up-to-date nationwide population-based mortality data collected across 40 years to examine secular trends in SSc mortality. The layout of data was clear and structured. A brief summary of the general management of SSc may fill in the gap of the study results and make the discussion easier to digest. Moreover, age- and sex-specific trends of SSc-ASMR were also examined to help researchers identify particular demographic groups whose APC in mortality trends remain suboptimal. 

However, several limitations and caveats should be highlighted. First, the mortality rates calculated in the cohort studies were actually case fatality rates, with the number of individuals diagnosed with SSc serving as the denominator. Due to the fact that the denominator of the mortality rate in the present study was the whole population, the rate was influenced by two factors: the incidence (prevalence) rate and the case fatality rate. Second, several clinical data, including the year of diagnosis, severity of SSc, and medications used, were not available in this study. Subsequently, further analysis based on the factors related to mortality was unavailable. Third, due to the diverse clinical presentation of SSc, the condition may be omitted from the death certificates in 30–50% of SSc-related deaths caused by renal crisis, PAH, or malignancy. Additionally, validation of the findings relies on the difficult-to-determine accuracy of physicians’ coding of causes of death on the death certificates. Nonetheless, it is known that mortality information on death certificates frequently underestimates the mortality burden associated with autoimmune diseases. It has been suggested that this bias is likely to be a persistent issue [28]. Furthermore, the purpose of this study is to examine mortality trends; thus, the underreporting rate is unlikely to systematically bias the results over time in the absence of actions targeting certifying practices. Lastly, racial and geographic disparities in SSc mortality rates were beyond the scope of this study, thus further investigation will be addressed in the future.

## 5. Conclusions

In conclusion, the MCD of SSc had 1.48 times the number of deaths that the UCD of SSc had in the past four decades, and the overall ratio of UCD to MCD gradually increased over time. Given the overall mortality of SSc steadily declined in the past two decades, the mortality RR of female over male decedents increased. That is, the sex disparity in SSc morality persisted and became more prominent, and development of therapeutic therapy is urgent. Thanks to widely used medications for complications of SSc, the age- and sex-adjusted mortality rates were decreasing. Additional research may be required to determine the causes of the suboptimal trends. Further evaluation of the contributing cause of death in SSc is also suggested.

## Figures and Tables

**Figure 1 ijerph-19-15088-f001:**
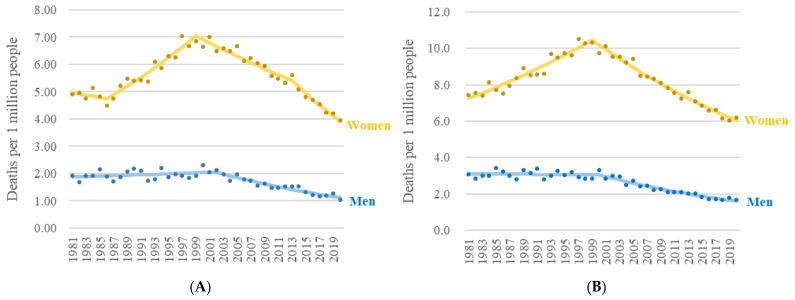
Age-standardized mortality rate of systemic sclerosis according to underlying causes of death (UCD) and multiple causes of death (MCD) stratified by sex in the United States, 1981–2020. (**A**) Overall, UCD. (**B**) Overall, MCD. (**C**) Age 0–44, UCD. (**D**) Age 0–44, MCD. (**E**) Age 45–64, UCD. (**F**) Age 45–64, MCD. (**G**) Age 65–74, UCD. (**H**) Age 65–74, MCD. (**I**) Age ≥75, UCD. (**J**) Age ≥75, MCD.

**Figure 2 ijerph-19-15088-f002:**
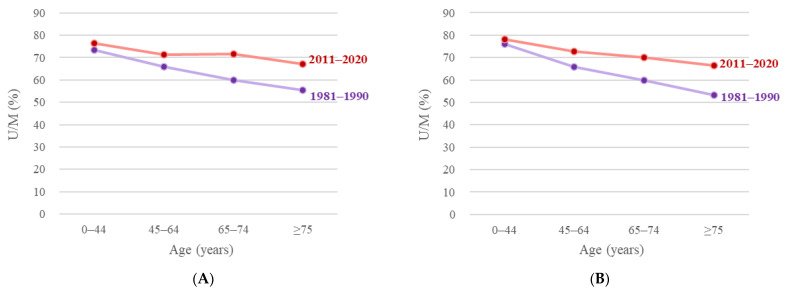
Proportion of number of deaths according to underlying cause of death (U) among those according to multiple causes of death (M) in systemic sclerosis mortality by sex in the United States, 1981–1990 versus 2011–2020. (**A**) Men. (**B**) Women.

**Table 1 ijerph-19-15088-t001:** Number of deaths (No), age-standardized mortality rate (ASMR), and mortality rate ratio (RR) of systemic sclerosis stratified by sex and age according to the underlying cause of death (UCD) and multiple causes of death (MCD), 1981–1990 versus 2011–2020.

	UCD								MCD							
	(1) 1981–1990			(2) 2011–2020			(2)/(1)		(1) 1981–1990			(2) 2011–2020			(2)/(1)	
Age	No	%	ASMR	No	%	ASMR	RR	(95% CI)	No	%	ASMR	No	%	ASMR	RR	(95% CI)
Both sexes																
Overall *	8067	100.0	3.63	11,934	100.0	3.20	0.88	(0.86–0.91)	12,896	100.0	5.81	16,965	100.0	4.54	0.78	(0.76–0.80)
0–44	1065	13.2	0.64	859	7.2	0.45	0.70	(0.64–0.77)	1411	10.9	0.85	1106	6.5	0.58	0.68	(0.63–0.74)
45–64	3222	39.9	7.16	4040	33.9	4.84	0.68	(0.64–0.71)	4899	38.0	10.89	5572	32.8	6.67	0.61	(0.59–0.64)
65–74	2316	28.7	13.62	3401	28.5	12.21	0.90	(0.85–0.95)	3863	30.0	22.71	4835	28.5	17.36	0.76	(0.73–0.80)
≥75	1464	18.1	12.49	3634	30.5	17.55	1.41	(1.32–1.49)	2723	21.1	23.22	5452	32.1	26.33	1.13	(1.08–1.19)
Men																
Overall *	1914	100.0	1.94	2294	100.0	1.32	0.68	(0.64–0.72)	3025	100.0	3.09	3229	100.0	1.86	0.60	(0.57–0.63)
0–44	237	12.4	0.28	222	9.7	0.23	0.81	(0.68–0.98)	323	10.7	0.39	290	9.0	0.30	0.78	(0.66–0.91)
45–64	866	45.2	4.03	974	42.5	2.39	0.59	(0.54–0.65)	1315	43.5	6.12	1366	42.3	3.35	0.55	(0.51–0.59)
65–74	539	28.2	7.27	656	28.6	5.04	0.69	(0.62–0.78)	897	29.7	12.10	914	28.3	7.02	0.58	(0.53–0.64)
≥75	272	14.2	6.60	442	19.3	5.25	0.80	(0.68–0.93)	490	16.2	11.88	659	20.4	7.82	0.66	(0.59–0.74)
Women																
Overall *	6153	100.0	5.01	9640	100.0	4.77	0.95	(0.92–0.98)	9871	100.0	7.98	13,736	100.0	6.77	0.85	(0.83–0.87)
0–44	828	13.5	1.01	637	6.6	0.68	0.68	(0.61–0.75)	1088	11.0	1.32	816	5.9	0.87	0.66	(0.60–0.72)
45–64	2356	38.3	10.04	3066	31.8	7.17	0.71	(0.68–0.75)	3584	36.3	15.27	4206	30.6	9.84	0.64	(0.62–0.67)
65–74	1777	28.9	18.52	2745	28.5	18.50	1.00	(0.94–1.06)	2966	30.0	30.91	3921	28.5	26.43	0.86	(0.82–0.90)
≥75	1192	19.4	15.68	3192	33.1	26.00	1.66	(1.55–1.77)	2233	22.6	29.37	4793	34.9	39.04	1.33	(1.26–1.40)

* Mortality rate was age-adjusted for the United States population in 2000 as standard population.

**Table 2 ijerph-19-15088-t002:** Summary of general management for systemic sclerosis.

SSc-Related Complication	Recommended Treatment
Heart failure	Standard heart failure therapy Immunomodulators: consider for myocarditis
Interstitial lung disease	MMF Cyclophosphamide TKI (Nintedanib) Lung transplantation: severe case * CD20 monoclonal antibody (Rituximab)
Pulmonary hypertension	PDE5i (Sildenafil, Tadalafil) ERA (Ambrisentan, Bosentan, Macitentan) Prostacyclin pathway agonists (Epoprostenol): severe case Lung transplantation: severe case
Gastrointestinal diseases	PPI: GERD, esophagitis, gastritis, ulcer, strictures Prokinetic agents: GERD, dysphagia Cyclical antibiotics: malabsorption secondary to bacterial overgrowth Anti-diarrheal agents: fecal incontinence resulting from anorectal diseases
Scleroderma renal crisis	ACEi + anti-hypertensive agents Dialysis Kidney transplantation: severe case
Skin involvement	Methotrexate MMF Cyclophosphamide * CD20 monoclonal antibody (Rituximab)
Raynaud’s phenomenon	CCB (Amlodipine, Nifedipine) ARB Prostanoid (Iloprost): severe case
Digital ulcer	Maximize therapy for Raynaud’s phenomenon

* Rituximab is shown to be effective and safe for SSc-ILD and SSc-related skin presentation, although it is currently approved for SSc only in Japan and not in the United States or Europe [10,11]. Therefore, it was not yet recommended by EULAR Committees but was listed in the table for information. ACEi, angiotensin-converting enzyme inhibitor; ARB, angiotensin II receptor antagonist; CCB, calcium channel blocker; ERA, endothelin I receptor antagonist; GERD, gastroesophageal reflux disease; MMF; mycophenolate mofetil; PDE5i, phosphodiesterase type 5 inhibitor; PPI, proton pump inhibitor; SSc, systemic sclerosis; TKI, tyrosine kinase inhibitor.

## Data Availability

Data are available from the corresponding author upon reasonable request.

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
