# Peer review of "Secular Trends in Systemic Sclerosis Mortality in the United States from 1981 to 2020"

_ijerph, 2022, doi:10.3390/ijerph192215088_

Round 1
Reviewer 1 Report
This study will analyze SSc mortality based on data regarding underlying cause of death (UCD) and multiple causes of death (MCD) over the past four decades.
Comments
This manuscript reports that The SSc mortality rates in all the–sex groups significantly decreased over the past two decades, which indicates a significant improvement in the efficacy of the treatments used and in the health care provided to these patients. The manuscript honestly points out the possible limitations of the study, which are inherent to the applied methodology itself.
The collected data are well analyzed statistically, are clearly explained and are well represented in the figures of the manuscript. For this reason, this manuscript offers very relevant information
Reviewer 2 Report
Systemic sclerosis is a serious disease, and the presentation over historical statistical data is highly valuable.
However, there is a need for improvements in the presentation of the results.
Page 2
Introduction, page 2 paragraph 3
* UCD - spell out first time in the main text
Method, page 2, first paragraph
* This is not a complete sentence: "Using the NCHS's multiple-cause mortality files to identify all deaths in the United States from 1981 through 2020 in which SSc was mentioned on the death certificate"
Result section
The whole result section is presented as unnecessary technical and difficult to read. The main picture as seen from the figures is drowning in technicalities.
It appears that too much focus is paid to highlighting the statistics and the presentation of agreement and disagreement between different data so that the easy presentation of the main content is lost.
For example, it would be useful to present first the figures and in the text present the general curvature that is visible in the figures before presenting how the data are split into different time periods.
Tables 1 and 2 are difficult to read as the lines are packed. As the tables are difficult to read, and also the text is very technical, it is difficult to get an overview. The figures, which are clear and easy to view, come after "the heavy technical presentation" of tables 1 and 2. The readers, therefore, easily get lost before reaching the figures.
In the discussion, medical treatments such as Rituximab, are mentioned. But details on when it was used for SSc are not presented. Could it be possible to give as suppl material in the results section, a presentation of common medical treatment of SSc over the time period that is covered in the study?
Do not refer to Figures in the discussion. Focus on presenting in clear words the information and discussing the findings.
Reviewer 3 Report
In this study, the Author investigated the trend of SSc mortality in the United States in the last decades, showing that SSc mortality rates in all the age–sex groups significantly decreased over the past two decades.
Overall, the study is well conducted and written. The discussion is fine and acknowledgement of several potential limitation is present. The conclusion is supported by the results presented. Figures and Tables are informative.
Tables are hard to read with the current layout.
English is fine and needs minor spell checks.
Round 2
Reviewer 2 Report
The present manuscript has high quality, reports important information, and is easy to read.